# Efficacy and Safety of Neoadjuvant Gemcitabine Plus Nab-Paclitaxel in Borderline Resectable and Locally Advanced Pancreatic Cancer—A Systematic Review and Meta-Analysis

**DOI:** 10.3390/cancers13174326

**Published:** 2021-08-27

**Authors:** Marko Damm, Ljupcho Efremov, Benedikt Birnbach, Gretel Terrero, Jörg Kleeff, Rafael Mikolajczyk, Jonas Rosendahl, Patrick Michl, Sebastian Krug

**Affiliations:** 1Department of Internal Medicine I, University Hospital Halle, Martin-Luther-University Halle-Wittenberg, D-06120 Halle (Saale), Germany; marko.damm@uk-halle.de (M.D.); jonas.rosendahl@uk-halle.de (J.R.); sebastian.krug@uk-halle.de (S.K.); 2Institute for Medical Epidemiology, Biometrics and Informatics (IMEBI), Interdisciplinary Center for Health Sciences, Martin-Luther-University Halle-Wittenberg, D-06112 Halle (Saale), Germany; ljupcho.efremov@uk-halle.de (L.E.); b.birnbach@gmx.de (B.B.); rafael.mikolajczyk@uk-halle.de (R.M.); 3Department of Radiation Oncology, Martin-Luther-University Halle-Wittenberg, D-06120 Halle (Saale), Germany; 4Department of Medicine, Sylvester Comprehensive Cancer Center, University of Miami, Miami, FL 33136, USA; gretel.terrero@jhsmiami.org; 5Department of Surgery, University Hospital Halle, Martin-Luther-University Halle-Wittenberg, D-06120 Halle (Saale), Germany; joerg.kleeff@uk-halle.de

**Keywords:** pancreatic cancer, neoadjuvant, resection, borderline resectable, locally advanced, albumin-bound paclitaxel, gemcitabine

## Abstract

**Simple Summary:**

Due to the availability of effective combination chemotherapies such as gemcitabine/nab-paclitaxel (GNP) or FOLFIRINOX, neoadjuvant treatment of borderline resectable (BRPC) and locally advanced pancreatic cancer (LAPC) has been increasingly investigated in recent years. However, due to toxicity, FOLFIRINOX is only available for selected patients and data on GNP are scarce. The aim of this systematic review and meta-analysis, which is to our knowledge the first addressing this question, is to evaluate the value of GNP in patients with BRPC and LAPC. We provide a comprehensive overview on data of 21 studies, comprising 950 patients treated with neoadjuvant GNP. The pooled overall and R0 resection rates were 36% and 26%, respectively. Resection rates were higher in BRPC (49%) compared to LAPC (16%). With acceptable toxicity and a median overall survival rate ranging from 12 to 30 months, neoadjuvant GNP has considerable value in this setting, with more prospective trials being warranted.

**Abstract:**

Therapy with gemcitabine and nab-paclitaxel (GNP) is the most commonly used palliative chemotherapy, but its advantage in the neoadjuvant setting remains unclear. Accordingly, our aim is to evaluate the impact of first-line neoadjuvant therapy with GNP in patients with borderline resectable (BRPC) and locally advanced pancreatic cancer (LAPC). A systematic search for published studies until August 2020 was performed. The primary endpoint included resection and R0 resection rates in the intention-to-treat population. Secondary endpoints were response rate, survival and toxicity. Among 21 studies, 950 patients who received neoadjuvant GNP were evaluated. Treatment with GNP resulted in surgical resection and R0 resection rates as follows: 49% (95% CI 30–68%) and 36% (95% CI 17–58%) for BRPC and 16% (95% CI 7–26%) and 11% (95% CI 5–19%) for LAPC, respectively. The objective response rates and the median overall survival (mOS) ranged from 0 to 67% and 12 to 30 months, respectively. Neutropenia (range 5–77%) and neuropathy (range 0–22%) were the most commonly reported grade 3 to 4 adverse events. Neoadjuvant chemotherapy with GNP can be performed safely and with valuable effects in patients with BRPC and LAPC. The utility of GNP in comparison to FOLFIRINOX in the neoadjuvant setting requires further investigation in prospective randomized trials.

## 1. Introduction

The incidence of pancreatic ductal adenocarcinoma (PDAC) is steadily increasing in the Western world. Current estimates project a disease rate of 60,500 cases in the United States in 2021 [1]. Combination chemotherapy, such as gemcitabine together with nab-paclitaxel (GNP) or 5-fluorouracil combined with oxaliplatin and irinotecan (FOLFIRINOX), significantly improved overall survival in patients with PDAC in the adjuvant and/or metastatic setting [2,3,4]. However, despite this progress, the 5-year survival rate remains one of the worst among solid malignancies [1]. According to recent studies, PDAC is predicted to be the second leading cause of cancer-related death in Germany and the US by 2030, thus having a high socioeconomic impact on healthcare systems [5,6].

Resection is the only possible chance for a cure, but due to the lack of screening methods and early symptoms, surgery is only feasible in 15–20% of the cases. The remaining patients with PDAC are either borderline resectable (BRPC), locally advanced (LAPC) or metastatic. After successful resection, adjuvant therapy has been shown to improve survival. A better understanding of therapy selection, initiation of therapy after resection and supportive options such as enzyme replacement and nutritional therapy resulted in increased median survival of up to 4.5 years in selected patient groups [4,7,8].

Compared to other gastrointestinal malignancies, neoadjuvant therapy (NAT) to date is of limited value in PDAC. Nevertheless, there is an important rationale for NAT in BRPC and LAPC to increase the chance for complete resection and decrease the risk of local or systemic recurrence by local downstaging and elimination of micrometastases. In addition, NAT enables identification of patients with aggressive tumor biology despite systemic therapy who do not benefit from surgery [9]. In accordance with these considerations, previous studies suggested that NAT provides oncological benefits compared to upfront surgery in patients with BRPC [10,11]. Thus, NAT is the recommended therapy in BRPC according to the National Comprehensive Cancer Network (NCCN) and European Society of Medical Oncology (ESMO) guidelines [9,12,13].

With regard to combination chemotherapy in the neoadjuvant setting, most studies were performed with FOLFIRINOX [9]. Meta-analyses showed that neoadjuvant FOLFIRINOX resulted in a pooled resection rate of up to 68% and 26% in patients with BRPC and LAPC, respectively [14,15,16]). The median overall survival (mOS) across the studies varied between 11 and 34.2 (BRPC) and 10 and 32.7 months (LAPC) in the intention-to-treat population. In a highly selected cohort of LAPC patients, resection rates of up to 60% were described [17]. In addition, the mOS of resected BRPC/LAPC patients treated with neoadjuvant FOLFIRINOX together with radiotherapy reached 57.8 months in the study of Pietrasz and colleagues [18].

Most studies to date have indicated greater efficacy and associated longer survival for FOLFIRINOX compared to gemcitabine +/− nab-paclitaxel [2,4]. However, there are relevant limitations to these data: in the real-word setting outside of clinical trials, therapy with gemcitabine and nab-paclitaxel was shown to be non-inferior to modified FOLFIRINOX (mFOLFIRINOX) in the palliative setting [19,20,21]. Treatment with mFOLFIRINOX is only suitable for patients with an excellent performance status without relevant comorbidities, in contrast to GNP which was successfully administered in patients up to ECOG 2 with acceptable toxicity [2]. Unfortunately, predictive and clinically applicable markers for therapy stratification do not exist yet. For therapy with gemcitabine and nab-paclitaxel, the strongest reduction in mortality was observed for patients with liver metastases, more than three metastatic sites, a Karnofsky index of 70–80% or a significantly elevated CA19-9 [2]. In the palliative setting, the use of gemcitabine with nab-paclitaxel is therefore administered in up to 40–50% of cases, whereas mFOLFIRINOX is only used in 20–25% [19,20,21]. However, the number of randomized and prospective studies on perioperative or neoadjuvant approaches for any of these two systemic therapies is limited.

In the past few years, several studies investigating the efficacy and safety of GNP in the neoadjuvant setting have been published and the body of evidence is growing. However, no systematic analysis of existing data has been performed yet. To provide a comprehensive overview of current evidence, we performed a systematic review and meta-analysis of published literature to evaluate the overall resection rate, R0 resection rate, toxicity and survival outcomes after neoadjuvant chemotherapy with GNP in patients with BRPC or LAPC.

## 2. Materials and Methods

### 2.1. Search Strategy

A systematic search for eligible articles was performed in PubMed/MEDLINE, Cochrane Library/Cochrane Central Register of Controlled Trials/CENTRAL, Google Scholar, Web of Science and LIVIVO for published studies either in English or German until August 3rd, 2020. The search terms were “Pancreas”, “Cancer”, “Nab-paclitaxel”, “Gemcitabine” and relevant variants. The study protocol has been registered at the International Prospective Register of Systematic Reviews (PROSPERO CRD42019135326).

### 2.2. Inclusion and Exclusion Criteria

We included only randomized controlled trials and studies with prospective or retrospective observational design. Duplicates, case reports, letters, reviews and studies with no details on resection rates or studies including minors were excluded. We screened the title and abstract for each retrieved study for eligibility. If abstract screening indicated relevant content for the research question, the full text was further assessed. Studies reporting the use of first-line GNP as a neoadjuvant treatment in BRPC or LAPC with the intention to perform a resection of the tumor regardless of subsequent other treatments were selected. Studies that investigated the combination of GNP with other chemotherapy agents or radiotherapy in a neoadjuvant setting were also eligible. The reference lists of all included articles were manually searched for the identification of potentially missed studies. Preferred Reporting Items for Systematic Reviews and Meta-analysis (PRISMA) guidelines were followed when reporting results [22]. The flowchart for study selection criteria is shown in Figure 1.

### 2.3. Data Extraction

Two reviewers (M.D. and B.B.) independently screened articles to determine eligibility. If there were disagreements, a third reviewer (R.M.) was consulted. The following information was extracted: first author, year of publication, country, study design, study population (total number of patients analyzed), patient groups (median age, performance status, sex), tumor stage (location and local extent of the disease), diagnostic work-up (CT, MRI, laparoscopy for staging), type of intervention (treatment regimen and number of administered cycles, percentage of (chemo-) radiation, type and percentage of additional chemotherapy agents), surgical resection rates, R0 resection rates, duration of follow-up, overall survival (OS), progression-free survival (PFS), objective response rate (ORR) and grade 3/4 adverse events. Corresponding authors were contacted when data were missing or could not be extracted from the article. The primary outcomes were overall resection rates and R0 resection rates after first-line neoadjuvant chemotherapy with GNP. Additional outcomes were median PFS, median OS, ORR and rate of G3/G4 toxicity.

### 2.4. Statistical Analysis

The meta-analyses were performed using STATA 15.0 statistical software (StataCorp, 4905 Lakeway Drive, College Station, TX, USA) using the package metaprop [23]. Pooled estimates of proportions with corresponding 95% confidence intervals (CI) are reported, calculated using the Freeman–Tukey double arcsine transformation. Due to heterogeneity among studies, a random effects model was used. Heterogeneity was quantified by the chi-square and *I*^2^ test, significant when *p* < 0.05. Publication bias was explored using funnel plots and symmetry of the funnel plot was analyzed with visual inspection. Quality assessment of studies was performed using the Newcastle-Ottawa Scale (NOS) [24] and the Cochrane Collaboration’s tool for assessing risk of bias in randomized trials [25]. Two reviewers (L.E. and M.D.) performed the assessment independently and disagreements were discussed afterwards.

## 3. Results

### 3.1. Study Selection and Characteristics

Among 2131 identified records, 77 full-text articles were assessed for eligibility and 21 studies were included in the qualitative and quantitative synthesis [26,27,28,29,30,31,32,33,34,35,36,37,38,39,40,41,42,43,44,45,46]. Reasons for study exclusion are shown in the PRISMA flow diagram (Figure 1).

In total, the 21 studies involved 2570 patients undergoing different neoadjuvant chemotherapy regimens in patients with resectable, borderline resectable and locally advanced PDAC (Table 1).

Nine studies had a prospective design and 12 were retrospective cohort studies. The prospective studies comprised one phase 2 single-center randomized open-label study with two treatment arms [40], one phase 2 multicenter open-label single-arm [38], one prospective observational study [33] and six phase I studies [31,32,36,39,41].

Resectability status was based on NCCN criteria in 17 out of 21 studies. Criteria of MD Anderson Cancer Center (MDACC) [29] and American Hepato-Pancreato-Biliary Association/Society of Surgical Oncology/Society for Surgery of the Alimentary Tract (AHPBA/SSO/SSAT) were used in one study [38], respectively. Another study followed the Alliance classification [32] and one study did not provide information on how resectability status was defined [43].

Most of the studies (19/21) were intention-to-treat analyses, whereas two studies just included resected patients [27,32].

### 3.2. Results of Studies

For qualitative synthesis, patients with neoadjuvant therapies other than GNP (*n* = 1360), no neoadjuvant therapy (*n* = 173), patients with resectable PDAC (if data were reported separately from BRPC/LAPC, *n* = 70) and patients who were included in more than one study (Maggino et al. [33] and Weniger et al. [45], *n* = 17) were excluded.

Overall, 950 patients who received first-line neoadjuvant gemcitabine plus nab-paclitaxel were included in qualitative synthesis (Table 2). Among them, 401 (42%) were diagnosed with BRPC, 458 (48%) with LAPC and 78 (8%) were resectable, whereas 13 (1%) were not specified (BRPC or LAPC). The median age of the study populations ranged from 58 to 71 years.

#### 3.2.1. Treatment

In most studies (16/21), application of neoadjuvant GNP followed the standard protocol, which consisted of gemcitabine 1000 mg/m^2^ and nab-paclitaxel 125 mg/m^2^ at day 1, 8 and 15 whereas in three phase I studies different dose levels were investigated [31,39,46]. In one study, 81% of the patients received biweekly GNP [29] and one study did not report details on the chemotherapy protocol [45]. The median number of cycles administered ranged from 2 to 8, with four studies not providing detailed information. In 12/21 studies, a subset of patients ranging from 10 to 100% received neoadjuvant radiotherapy in addition to GNP. In one study, 14% (*n* = 3) received additional FOLFIRINOX [45], and additional cisplatin and capecitabine (PAXG protocol) were applied in 48–100% of the patients (*n* = 50) in two studies [39,40], and all patients (100%, *n* = 16) received S1 in the study of Kondo et al. [31]. Patients in five studies did not receive any additional neoadjuvant therapy (*n* = 277, refs. [27,34,36,38,43]).

#### 3.2.2. Response Rate

Radiological response after neoadjuvant therapy according to RECIST was available in 11/21 studies. The objective response rate (ORR) and disease control rate (DCR) ranged from 0 to 67% and from 57 to 100%, respectively. Of note, no patient showed complete response (CR) after therapy with GNP.

#### 3.2.3. Survival

The median overall survival (mOS) was reported in 13/21 studies and ranged from 12 to 29.9 months in studies with an ITT population. The mOS ranged from 19.8 to 43.6 months for patients undergoing secondary resection and from 10.2 to 16 months for non-resected patients. In six studies, the median progression-free survival (mPFS) was reported as ranging from 8.2 to 14.8 months, while 15 studies did not provide PFS information. Six studies reported data on mOS separately for BRPC or LAPC. The mOS in ITT analysis for patients with BRPC ranged from 14.5 to 27.9 and with LAPC from 15.7 to 19.9 months.

#### 3.2.4. Toxicity

Only 8/21 studies reported data on the toxicity of neoadjuvant GNP. The overall incidence of G3 and G4 adverse events ranged from 5 to 90% (Table 2). Data on grade ≥3 neutropenia and neuropathy were most commonly reported (14/21), ranging from 5 to 77% and from 0 to 22%, respectively (Appendix A).

### 3.3. Synthesis of Studies

#### 3.3.1. Resection Rates

For quantitative synthesis, 19 studies comprising 739 patients treated with neoadjuvant GNP were eligible. Two studies comprising 211 cases that only included resected patients were excluded [27,32]. Of 739 patients, 282 (38%) were diagnosed with BRPC and 444 (60%) with LAPC, whereas 13 (2%) were not specified [41].

After a median of 2–8 cycles of first-line neoadjuvant GNP, 36% (95% CI 24–49%) of the patients underwent surgical resection. In 26% (95% CI 15–38%) of all patients who underwent treatment, R0 resection was achieved (Figure 2). Of note, 30% (*n* = 224) received additional radiotherapy and 9% received additional chemotherapy (S1/FOLFIRINOX/cisplatin/capecitabine, *n* = 69). Pooled proportions of prospective studies (*n* = 396) showed an overall resection rate of 42% (95% CI 24–61%) and an R0 resection rate of 22% (95% CI 10–36%) (Appendix A).

Among 282 patients with BRPC, surgical resection and R0 resection rates were 49% (95% CI 30–68%) and 36% (95% CI 17–58%), respectively (Figure 2). Patients with LAPC (*n* = 444) showed overall resection and R0 resection rates of 16% (95% CI 7–26%) and 11% (95% CI 5–19%). Based on resected patients, the pooled R0 resection rates were 85% (95% CI 68–97%), 89% (95% CI 69–100%) and 77% (95% CI 51–97%) for all patients, BRPC and LAPC, respectively. In prospective studies, R0 resection was achieved in 74% (95% CI 55–90%) of the resected patients.

#### 3.3.2. Resection Margin

Of note, there was heterogeneity of R0 definition among the studies: 9 of 19 studies provided information on definition of R0 resection. In six studies, absence of tumor at the margin (UICC definition), and in three studies, a minimum distance between tumor and margin of >1 mm (College of American Pathologists (CAP) definition) were defined as R0 resection (Table 2).

### 3.4. Quality Assessment

Overall, the systematic review comprised 13 retrospective cohort studies and 8 clinical trials. The retrospective cohort studies were assessed for quality using the Newcastle-Ottawa scale (NOS). Our analysis assessed four studies as having good quality, eight as having moderate quality and one was assessed as a poor-quality study (Appendix A). Using the Cochrane Collaboration’s tool for assessing risk of bias, seven of the clinical trials were assessed to have high risk of bias, due to their study design, having no randomization performed and being open-label (Appendix A). One RCT was assessed as having unclear risk of bias, since it performed randomization of participants in the two study arms, but there was no blinding of either patients or researchers. Visual analysis of the funnel plot revealed, in general, symmetry around the summary proportion, which indicates absence of publication bias (Appendix A).

## 4. Discussion

To our knowledge, this is the first systematic review and meta-analysis specifically addressing outcomes of neoadjuvant gemcitabine and nab-paclitaxel (GNP) in patients with BRPC or LAPC. With a pooled overall resection rate of 36% (739 patients, 19 studies), a pooled R0 resection rate of 26% (631 patients, 16 studies) and a median overall survival rate ranging from 12 to 30 months in ITT analysis, our data indicate that neoadjuvant treatment with GNP has considerable beneficial value.

In many other solid tumors of the gastrointestinal tract such as esophageal, gastric and rectal cancers, neoadjuvant or perioperative therapies represent an established therapeutic concept. In PDAC, however, this approach has not been implemented as standard treatment. So far, the standard in resectable disease is surgery with macroscopic complete resection followed by standard-of-care adjuvant therapy. In the non-metastatic setting, resection and in particular R0 resection are the ultimate goals since both are major determinants of long-term survival [47,48]. However, the assessment of resectability in localized PDAC represents a clinical challenge. Radiologic techniques alone are unable to reliably stratify the local disease stage and distinguish between BRPC and LAPC [49]. In addition, there is still no consensus in international guidelines on the management of BRPC/LAPC. Therefore, the management frequently depends on the local surgical expertise and assessment of the multidisciplinary board [49,50].

In the present meta-analysis, over 80% of the included studies assessed resectability status according to the NCCN guidelines, indicating that these recommendations have now been widely accepted [12]. Despite limited evidence regarding the optimal therapeutic regimen, neoadjuvant therapy is recommended in these guidelines for BRPC and should also be considered in PDAC patients who are resectable by imaging but show high-risk features such as very high CA 19-9, large primary tumors or lymph nodes and excessive weight loss or extreme pain. For selected patients with LAPC and good performance status, induction chemotherapy for 4–6 months or chemoradiation are often used, although evidence based on large randomized clinical trials is still lacking. If there is at least radiographic stability and marked clinical improvement or decline in CA 19-9, evaluation for surgery in a high-volume center is recommended [12].

In our meta-analysis, as expected, the overall and R0 resection rates were higher in patients with BRPC compared to patients with LAPC. However, they were significantly lower than in the most recently published meta-analyses by Janssen and Chen [14,16]. Sixty-eight percent of patients with BRPC and 26% of patients with LAPC underwent resection after neoadjuvant therapy with FOLFIRINOX. In 84% and 88% of resected patients, R0 resections could be achieved, respectively. In line with these results are the findings of Xu et al., who analyzed 958 patients with BRPC/LAPC and treatment with neoadjuvant FOLFIRINOX in their meta-analysis [51]. The pooled overall resection rate and R0 resection rate with 55% and 40%, respectively, were also higher than in our study. However, these three meta-analyses on FOLFIRINOX used partly overlapping study populations. Because of the probability of bias, direct comparability of resection rates between the different meta-analyses is difficult.

First, in parallel with studies in the palliative setting, it must be assumed that the application of FOLFIRINOX is associated with a strong patient selection bias, especially in retrospective cohort studies. On average, patients receiving FOLFIRINOX are 10 years younger than the average age of the PDAC population [2,3]. Therefore, only fewer than 25% of patients with LAPC or metastatic PDAC receive systemic therapy with FOLFIRINOX in clinical real-world data [19]. Nevertheless, in the neoadjuvant setting, there are limited data.

Second, the exact protocols and the duration of neoadjuvant chemotherapies, as well as the proportion of additional (chemo-) radiation (CRT) performed, varied considerably between the studies. In the meta-analyses of Chen, Xu and Janssen et al., the median number of neoadjuvant FOLFIRINOX cycles ranged from 3 to 11.5 [14,16,51]. The median number of administered GNP cycles in studies evaluated in our meta-analysis also varied notably, ranging from 2 to 8 cycles. For FOLFIRINOX used in LAPC, subgroup analyses did not show significant differences in the rates of overall and R0 resection between the groups receiving less vs. more than six cycles [16]. Similarly, there was no difference between those groups in BRPC patients [14]. In the present analysis, only 24% of the studies (5/21) followed the standard protocol of GNP on d1, d8 and d15 in a 28-day cycle without dose modifications or additional neoadjuvant therapy. Likewise, in the meta-analysis of Xu et al., only 22% (5/23) followed the standard FOLFIRINOX protocol. In 70% of the studies included by Xu et al., additional radiotherapy was administered, and in 26%, FOLFIRINOX was modified, mostly by omission of the 5FU bolus or a reduced irinotecan dose. This is in line with the findings in the palliative setting, where no differences on radiological response were detected, but toxicities were significantly increased when the standard protocol compared to modified FOLFIRINOX was used [52].

Additional (chemo-) radiation is widely used in the neoadjuvant setting, as it was applied in 57% of the studies with GNP and in 70–71% of the studies with FOLFIRINOX [14,51]. Before 2009, radiotherapy was part of neoadjuvant treatment regimens in almost all studies [53]. The usefulness of local radiation in addition to systemic therapy continues to be controversial. The ESPAC-1 trial, which showed unfavorable effects of adjuvant chemoradiation compared to adjuvant chemotherapy, has led to a decreased use of neoadjuvant radiotherapy in most European centers [9]. In contrast, a recent retrospective multicenter cohort study of patients with BRPC/LAPC showed significantly higher R0 resection rates (89.2% vs. 76.3%), more pathologic major responses (33.3% vs. 12.9%) and longer OS (57.8 vs. 35.5 months) in the group of patients who received CRT in addition to FOLFIRINOX compared to those without CRT [18]. However, the LAP07 study, a large RCT comparing neoadjuvant chemotherapy with gemcitabine +/− erlotinib with chemoradiation in LAPC did not show a significant difference in OS [54]. In this trial, probably due to the less effective chemotherapy used, resection of the primary tumor was only achieved in 4%. Whether intensified chemotherapy with GNP or FOLFIRINOX in combination with radiation achieves benefits is being explored in large multicenter prospective trials, including the ongoing CONKO-007 trial (NCT01827553).

Third, different or unclear definitions of resectability or R0 resection in the studies impede the comparability of the results. While in our analysis most of the studies followed the NCCN guidelines, up to seven different classifications of resectability were used in a total of 24 studies in the meta-analysis of Janssen et al. [14]. In addition, for one-fourth of the studies in Xu et al., no information on resectability criteria was available [51]. Regarding the definition of R0 resection, many studies use the UICC definition in analogy to the circumferential resection margin (CRM) concept and distinguish “R0 narrow” (distance between tumor tissue and resection margin ≤ 1 mm) from “R0 wide“ (distance > 1 mm) and R1 (tumor reaching the margin). However, other studies and expert committees such as the College of American Pathologists (CAP) define R0 in analogy to UICC “R0 wide” (distance > 1 mm) and refer to R0 narrow as R1 [12]. Comparable to the variability of R0 definitions in other meta-analyses, 4 out of 21 studies used the CAP definition, 6 used the UICC definition of R0 and 11 studies did not provide information in the present meta-analysis [14]. However, evidence regarding the adequate margin in PDAC is currently scarce [12].

The recent NEOLAP study was published after our systemic literature search was completed and was therefore not included in our analysis. However, these data have provided additional insights into many open questions in the search for the optimal chemotherapy regime. In patients with LAPC, which was the target population, neoadjuvant therapy consisting of four cycles of GNP versus two cycles of GNP plus four cycles of FOLFIRINOX was investigated [55]. The sequence of both therapies was implemented based on the preclinical assumption that nab-paclitaxel depletes the pronounced stromal reaction and thereby improves the intratumoral efficacy of subsequent chemotherapeutic agents [56,57]. Of the 130 patients included and randomized, 63% received secondary surgical exploration, of which 35.9% were resected in the GNP arm and 43.9% in the sequential arm, which was not statistically significant. There was also no difference in ORR and DCR between both neoadjuvant treatment groups. The R0 resection rate was similar in both arms (65% vs. 69%), which was lower than reported by Chen and colleagues (88%) [16]. Although subgroup analyses for T and N stage of the NEOLAP trial included only small numbers of patients, there was a trend for tumor downsizing after sequential therapy.

The ultimate goal is to determine whether the effect of neoadjuvant therapy followed by resection also has a positive impact on overall survival. Irrespective of categorization into BRPC and LAPC, our meta-analysis showed strong heterogeneity with variations for mOS of 12–30 months for the entire cohort. Previous patient-level meta-analyses of ITT populations showed a mOS of 22 and 24 months for BRPC and LAPC treated with neoadjuvant FOLFIRINOX, respectively [14,15]. Interestingly, Reni et al. have shown that the outcome after neoadjuvant systemic therapy and successful resection did not vary between BRPC and LAPC [58].

The NEOLAP study achieved a mOS of 18.5 months in the GNP group and 20.7 months in the sequential GNP/FOLFIRINOX group, which was not significantly different and altogether lower than expected from previous studies [55]. However, the NEOLAP trial demonstrates that intensified therapy with FOLFIRINOX had no significant benefit when compared with four cycles of GNP treatment. For daily practice, this is a very important finding, as therapy with GNP is potentially a valid option for all patients, even with reduced general condition and comorbidities. Interestingly, survival of R1-resected versus unresected patients was similar at 16–17 months, while patients with R0 resection experience the most pronounced benefit from neoadjuvant therapy (mOS 40.2 months).

No data exist on the impact of adjuvant therapy after neoadjuvant pretreatment and resection. Two-thirds of the patients in the NEOLAP study received adjuvant therapy with no difference between the groups. Compared to historical data on adjuvant therapy alone after curative resection, neoadjuvant therapy does not affect capability to receive adjuvant therapy [59,60].

In our meta-analysis, 11/21 studies provided information on adjuvant therapy which was applied in 30–96% of the resected patients. Gemenetzis and colleagues showed that there was no significant difference in postoperative PFS or OS between patients with adjuvant therapy and patients without [28]. In contrast, Weniger et al. showed a trend towards improved mOS following adjuvant therapy compared to postoperative observation (47 vs. 30 months, *p* = 0.06) [45]. However, to clarify those questions, studies powered for a perioperative therapy concept for BRCP/LAPC are warranted.

In the present meta-analysis, the most commonly reported grade ≥ 3 adverse events for neoadjuvant GNP were neutropenia and neuropathy. The occurrence of febrile neutropenia was reported with ≤4% in most of the studies. No deaths were attributed to GNP. These results are comparable to the findings of the phase III landmark trial conducted in the palliative setting [2]. The rate of febrile neutropenia due to FOLFIRINOX was higher in studies without prophylactic G-CSF administration [14]. In comparison, in the prospective NEOLAP study, no substantial differences in toxicity between both treatment strategies were observed. Nonetheless, in the FOLFIRINOX arm, in one-fifth of patients, neoadjuvant therapy had to be terminated due to side effects. In addition, treatment interruptions and dose modifications were significantly more frequent than under GNP [55].

Some limitations of our analysis need to be considered. More than half of the included studies were retrospective, which increases the risk of selection bias. Furthermore, two-thirds of the trials with a prospective design were phase I studies with small sample sizes. Some studies also demonstrated a limited follow-up period, among which mOS data should be interpreted tentatively. In addition, quality assessment revealed that only 30% (4/13) of the retrospective studies had good quality, and almost all of the clinical trials had high risk of bias, which might impair the validity of the observed results. Although most of the studies followed NCCN criteria for resectability, there was substantial heterogeneity, probably due to differences in patient characteristics, duration and dosage of neoadjuvant chemotherapy, such as variations in therapy additives. Similarly, the interpretation of outcomes might be hampered in part due to the lack of important information, such as data regarding the extent and impact of adjuvant therapy or how R0 resection was defined.

Promising studies in the field of neoadjuvant therapy, such as NEONAX (adjuvant vs. perioperative GNP in resectable PDAC), CONCO-007 (neoadjuvant chemotherapy vs. chemoradiation in LAPC) and PREOPANC-2 (neoadjuvant FOLFIRINOX vs. chemoradiation in resectable PDAC and BRPC) are currently ongoing and are eagerly awaited. Moreover, a randomized phase III study comparing neoadjuvant GNP with modified FOLFIRINOX in patients with BRPC/LAPC (NCT0461782) with planned enrollment of 300 participants and estimated completion in 09/2023 will hopefully fill the current evidence gap on the role of GNP in BRPC and LAPC.

## 5. Conclusions

In conclusion, this systemic review and meta-analysis demonstrates the feasibility of neoadjuvant GNP in patients with BRPC or LAPC, representing a reasonable alternative in this setting, when comorbidities preclude the use of FOLFIRINOX. However, more data from randomized prospective trials are needed.

## Figures and Tables

**Figure 1 cancers-13-04326-f001:**
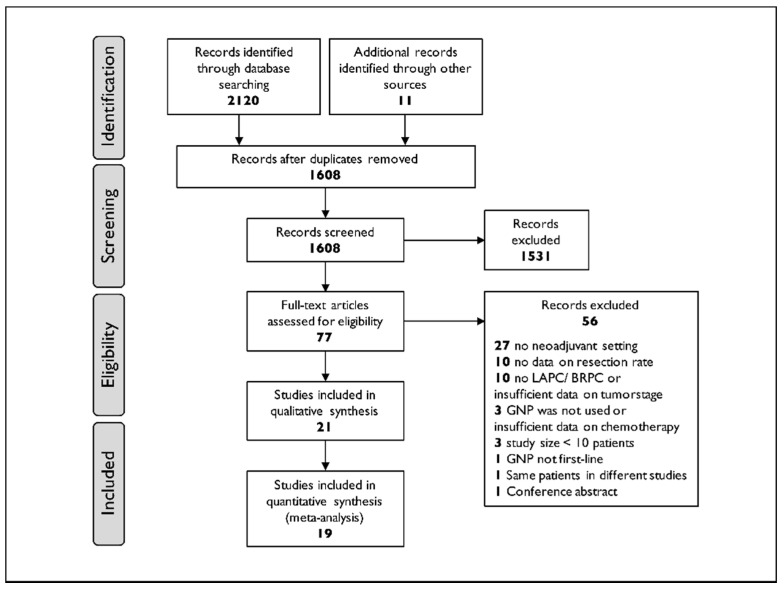
PRISMA flow diagram of the study selection process. After the database search, 1608 publications were screened and 21 studies met the criteria. Abbreviations: LAPC, locally advanced pancreatic cancer; BRPC, borderline resectable pancreatic cancer; GNP, gemcitabine and nab-paclitaxel.

**Figure 2 cancers-13-04326-f002:**
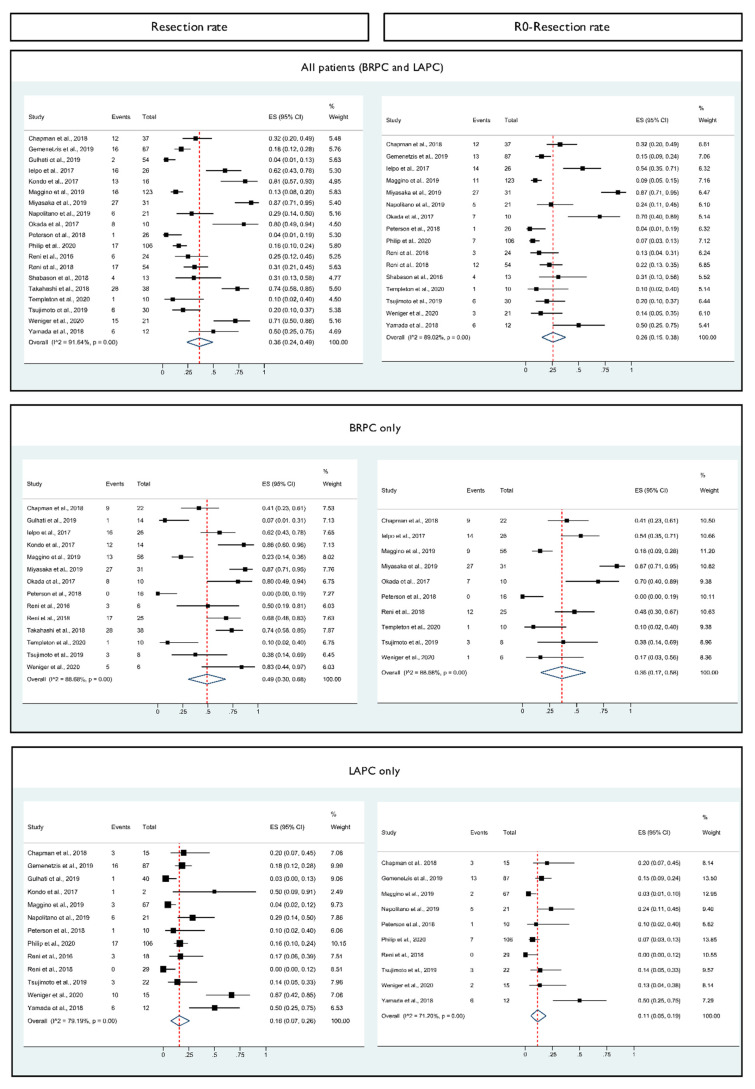
Forest plots showing the pooled proportions of resections and R0 resections (defined as absence of tumor at the margin or a minimum distance between tumor and margin of >1 mm) of all patients (*n* = 739, refs. [26,28,29,30,31,33,34,35,36,37,38,39,40,41,42,43,44,45,46]) with borderline resectable (BRPC) and locally advanced pancreatic cancer (LAPC), as such, separate analyses for patients with BRPC (*n* = 282, refs. [26,29,30,31,33,34,36,37,39,40,42,43,44,45]) or LAPC (*n* = 444, refs. [26,28,29,31,33,35,37,38,39,40,44,45,46]) only. Due to heterogeneity among studies, a random effects model was used. The proportions of the R0 resections were calculated on the basis of the total number of patients treated with neoadjuvant gemcitabine and nab-paclitaxel in the intention-to-treat population. The red dotted line and blue diamond shape indicate the overall pooled proportions (resection rate or R0 resection rate) including overall 95% confidence intervals (95% CI). The individual proportions of each study including 95% CI are shown in the column “ES (95% CI)”. The graphical representation corresponds to the black squares and lines whereby the size of the squares reflects the respective weighting in the analysis. Abbreviation: ES, effect size.

**Table 1 cancers-13-04326-t001:** Study characteristics including proportions of neoadjuvant treatment and resectability status [26,27,28,29,30,31,32,33,34,35,36,37,38,39,40,41,42,43,44,45,46].

No.	Study	Country	Study Period	Design	Definition Resectability	Resected Only	*n*	Neoadj. GNP	%	Neoadj. FFX	%	Neoadj. Other	%	Resectable	%	BRPC	%	LAPC	%
1	Chapman et al., 2018	US	2012–2016	retrospective, single-institutional	NCCN	No	120	37	*31*	83	*69*	0	*0*	0	*0*	79	*66*	41	*34*
2	Dhir et al., 2018	US	2011–2017	retrospective, single-institutional	NCCN	Yes	193	120	*62*	73	*38*	0	*0*	64	*33*	129	*67*	0	*0*
3	Gemenetzis et al., 2019	US	2013–2017	retrospective, single-institutional	NCCN	No	415	87	*21*	184	*44*	144	*35*	0	*0*	0	*0*	415	*100*
4	Gulhati et al., 2019	US	2013–2015	retrospective, single-institutional	MDACC	No	99	99	*100*	0	*0*	0	*0*	45	*46*	14	*14*	40	*40*
5	Ielpo et al., 2017	Spain	2007–2016	retrospective, single-institutional	NCCN	No	81	45	*56*	0	*0*	0	*0*	36	*44*	45	*56*	0	*0*
6	Kondo et al., 2017	Japan	2015–2016	Phase I, multicenter (*n* = 3)	NCCN	No	16	16	*100*	0	*0*	0	*0*	0	*0*	14	*88*	2	*12*
7	Macedo et al., 2019	US	2010–2016	retrospective, multicenter (*n* = 7)	Alliance	Yes	274	91	*33*	183	*67*	0	*0*	61	*22*	127	*46*	70	*26*
8	Maggino et al., 2019	Italy	2013–2015	prospective, single-institutional	NCCN, MDACC	No	680	123	*18*	260	*38*	187	*28*	0	*0*	267	*39*	413	*61*
9	Miyasaka et al., 2019	Japan	2010–2017	retrospective, single-institutional	NCCN	No	57	31	*54*	0	*0*	0	*0*	0	*0*	57	*100*	0	*0*
10	Napolitano et al., 2019	Italy	2014–2019	retrospective, single-institutional	NCCN	No	56	21	*38*	35	*63*	0	*0*	0	*0*	0	*0*	56	*100*
11	Okada et al., 2017	Japan	2015	Phase I, single-institutional	NCCN	No	10	10	*100*	0	*0*	0	*0*	0	*0*	10	*100*	0	*0*
12	Peterson et al., 2018	US	2013–2018	retrospective, multicenter (*n* = 2)	NCCN	No	32	32	*100*	0	*0*	0	*0*	6	*19*	16	*50*	10	*31*
13	Philip et al., 2020	Multinational	2015–2018	Phase II, multicenter (*n* = 35)	AHPBA/SSO/SSAT	No	107	106	*99*	0	*0*	0	*0*	0	*0*	0	*0*	107	*100*
14	Reni et al., 2016	Italy	2012–2014	Phase I, single-institutional	NCCN	No	24	24	*100*	0	*0*	0	*0*	0	*0*	6	*25*	18	*75*
15	Reni et al., 2018	Italy	2014–2016	Phase II, single-institutional	NCCN	No	54	54	*100*	0	*0*	0	*0*	0	*0*	25	*46*	29	*54*
16	Shabason et al., 2018	US	2014–2016	Phase I, NR	NCCN	No	13	13	*100*	0	*0*	0	*0*	0	*0*	4	*31*	5	*38*
17	Takahashi et al., 2018	Japan	NR	Phase I, single-institutional	NCCN	No	38	38	*100*	0	*0*	0	*0*	0	*0*	38	*100*	0	*0*
18	Templeton et al., 2020	Canada	2011–2017	retrospective, multicenter (NR)	NR	No	20	10	*50*	10	*50*	0	*0*	0	*0*	20	*100*	0	*0*
19	Tsujimoto et al., 2019	Japan	2015–2017	retrospective, single-institutional	NCCN	No	30	30	*100*	0	*0*	0	*0*	0	*0*	8	*27*	22	*73*
20	Weniger et al., 2020	Multinational	2011–2017	retrospective, multicenter (*n* = 7)	NCCN	No	239	38	*16*	135	*56*	66	*28*	0	*0*	98	*41*	141	*59*
21	Yamada et al., 2018	Japan	2016–2017	Phase I, single-institutional	NCCN	No	12	12	*100*	0	*0*	0	*0*	0	*0*	0	*0*	12	*100*

Abbreviations: NCCN, National Comprehensive Cancer Network; MDACC, MD Anderson Cancer Center; AHPBA, American Hepato-Pancreato-Biliary Association; SSO, Society of Surgical Oncology; SSAT, Society for Surgery of the Alimentary Tract; GNP, gemcitabine and nab-paclitaxel; FFX, FOLFIRINOX; BRPC, borderline resectable pancreatic cancer; LAPC, locally advanced pancreatic cancer; NR, not reported.

**Table 2 cancers-13-04326-t002:** Characteristics and outcomes of patients with neoadjuvant GNP [26,27,28,29,30,31,32,33,34,35,36,37,38,39,40,41,42,43,44,45,46].

No.	Study	*n*	Median Age	Resectable		BRPC		LAPC		GNP		Additional NAT			Resection		R0 Resection		Def. R0	Adjuvant Therapy		ORR%	mOS (mo)	mPFS (mo)	Median Follow-Up (mo)	Grade ≥3 Toxicity (%)
				*n*	%	*n*	%	*n*	%	Protocol	Median Cycles	Type	*n*	%	*n*	%	*n*	%		*n*	%					
1	Chapman et al., 2018	37	71	0	*0*	22	*59*	15	*41*	Standard *^4^	2	RTx	28	*76*	12	*32*	12	*32*	NR	8	*67*	8	19	8	16	15
2	Dhir et al., 2018 *^1^	120	69	49	*41*	71	*59*	0	*0*	Standard	2	-	0	*0*	NR	*-*	97	*-*	NR	90	*75*	NR	29	NR	22	NR
3	Gemenetzis et al., 2019 *^2^	87	65 *^3^	0	*0*	0	*0*	87	*100*	Standard	NR	RTx	50	*57*	16	*18*	13	*15*	>1 mm	NR	*30 *^5^*	NR	17–35 *^6^	NR	NR	NR
4	Gulhati et al., 2019	54	70 *^3^	0	*0*	14	*26*	40	*74*	Modified	NR	RTx	22	*41*	2	*4*	NR	*-*	NR	NR	*87 *^5^*	NR	NR	NR	NR	NR
5	Ielpo et al., 2017 *^2^	26	62 *^3^	0	*0*	26	*100*	0	*0*	Standard	>2	RTx	26	*100*	16	*62*	14	*54*	NR	NR	*61 *^5^*	NR	19	NR	NR	3
6	Kondo et al., 2017	16	67	0	*0*	14	*88*	2	*13*	Modified	6	S1	16	*100*	13	*81*	NR	*-*	NR	NR	*-*	31	NR	NR	NR	NR
7	Macedo et al., 2019 *^1^	91	66	29	*32*	48	*53*	14	*15*	Standard	3	RTx	34	*37*	NR	*-*	63	*-*	>1 mm	NR	*-*	NR	31	NR	NR	NR
8	Maggino et al., 2019 *^2^	123	NR	0	*0*	56	*46*	67	*54*	Standard	NR	RTx	23	*19*	16	*13*	11	*9*	>1 mm	NR	*-*	NR	NR	NR	NR	14
9	Miyasaka et al., 2019	31	68	0	*0*	31	*100*	0	*0*	Standard	3	-	0	*0*	27	*87*	27	*87*	NR	26	*96*	NR	28	NR	15	NR
10	Napolitano et al., 2019	21	65	0	*0*	0	*0*	21	*100*	Standard	5	RTx	2	*10*	6	*29*	5	*24*	NR	NR	*-*	33	16	NR	NR	5
11	Okada et al., 2017	10	70	0	*0*	10	*100*	0	*0*	Standard	2	-	0	*0*	8	*80*	7	*70*	≤1 mm	7	*88*	0	NR	NR	NR	9
12	Peterson et al., 2018 *^2^	26	70	0	*0*	16	*62*	10	*38*	Standard	3	RTx	10	*38*	1	*4*	1	*4*	>1 mm	NR	*-*	16 *^7^	12	NR	NR	NR
13	Philip et al., 2020	106	65	0	*0*	0	*0*	106	*100*	Standard	5	-	0	*0*	17	*16*	7	*7*	NR	NR	*-*	34	19	11	25	85
14	Reni et al., 2016	24	63	0	*0*	6	*25*	18	*75*	Modified	5	Cisplatin + Capecitabin	24	*100*	6	*25*	3	*13*	≤1 mm	NR	*-*	67	18	12	25	16
15	Reni et al., 2018	54	61–66	0	*0*	25	*46*	29	*54*	Modified	5	Cisplatin + Capecitabin	26	*48*	17	*31*	12	*22*	≤1 mm	NR	*-*	NR	19	10	31	NR
16	Shabason et al., 2018	13	58–63	0	*0*	NR	*NR*	NR	*NR*	Modified	2	RTx	9	*69*	4	*31*	4	*31*	≤1 mm	NR	*-*	NR	NR	NR	NR	NR
17	Takahashi et al., 2018	38	65	0	*0*	38	*100*	0	*0*	Modified	2	RTx	30	*79*	28	*74*	NR	*/*	≤1 mm	NR	*-*	67	NR	NR	NR	NR
18	Templeton et al., 2020	10	67	0	*0*	10	*100*	0	*0*	Standard	3	-	0	*0*	1	*10*	1	*10*	NR	1	*100*	NR	16	9	NR	NR
19	Tsujimoto et al., 2019	30	67	0	*0*	8	*27*	22	*73*	Standard	NR	RTx	12	*40*	6	*20*	6	*20*	NR	NR	*-*	49	30	15	25	NR
20	Weniger et al., 2020 *^2^	21	65 *^3^	0	*0*	6	*29*	15	*71*	NR	4	FFX	3	*14*	15	*71*	3	*14*	≤1 mm	12	*57*	38	NR	NR	NR	1
21	Yamada et al., 2018	12	61	0	*0*	0	*0*	12	*100*	Modified	8	RTx	12	*100*	6	*50*	6	*50*	NR	NR	*-*	42	NR	NR	10	NR

The percentages of the R0 resections were calculated on the basis of the total number of patients treated in the intention-to-treat (ITT) population. *^1^ no ITT analysis (only resected patients included), therefore no R0 resection rate was calculated; *^2^ the data were partially obtained through a separate request from the corresponding author of the respective study; *^3^ patients with treatments other than GNP included; *^4^ “Standard protocol” = gemcitabine 1000 mg/m^2^ and nab-paclitaxel 125 mg/m^2^ at d1, d8 and d15 in a 28-day cycle; *^5^ refers to all resected patients of the study (treatment other than GNP or patients with resectable PDAC included); *^6^ mOS 17 months in unresected, and 35 months in resected patients; *^7^ data including clinically borderline resectable patients (*n* = 6). Abbreviations: BRPC, borderline resectable pancreatic cancer; LAPC, locally advanced pancreatic cancer; GNP, gemcitabine and nab-paclitaxel; NAT, neoadjuvant therapy; Def., definition; ORR, objective response rate; mOS, median overall survival; mPFS, median progression-free survival; mo, months; RTx, radiotherapy; NR, not reported.

## Data Availability

The raw data presented in this study are available on request from the corresponding author.

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
