# Peer review of "Efficacy and Safety of Neoadjuvant Gemcitabine Plus Nab-Paclitaxel in Borderline Resectable and Locally Advanced Pancreatic Cancer—A Systematic Review and Meta-Analysis"

_cancers, 2021, doi:10.3390/cancers13174326_

Round 1

Reviewer 1 Report

This is a systematic review and a meta-analysis of neoadjuvant gemcitabine and nab-paclitaxel for borderline resectable and locally advanced pancreatic cancer. 

  1. Can you provide subgroup analysis of BR-PC and LA-PC?
  2. Some phase 1 studies were included and the follow up period was not long enough. Robustness of data on survival might be limited.
  3. Please include data on adjuvant therapy after surgical resection in the table. The adjuvant therapy after NAC is clinically important. Can the authors suggest its relevance through this systematic review?
  4. Appropriate duration of NAC has not been fixed. Can the authors provide relationship of NAC duration and resection rates?

Author Response

Reviewer 1:

This is a systematic review and a meta-analysis of neoadjuvant gemcitabine and nab-paclitaxel for borderline resectable and locally advanced pancreatic cancer.

Point 1:

  1. Can you provide subgroup analysis of BR-PC and LA-PC?

Response 1:

Thank you for this request. During the submission process, we discovered that we were unable to correctly integrate tables 1 and 2, which are of central importance for the manuscript, into the main document. These tables have disappeared into the appendix. We apologize for this error. Table 1 lists all studies separately according to the proportion of LAPC and BRPC. Table 2 then presents the primary endpoints resection rate, R0 resections, mPFS, mOS for both collectives.

Tables 1 and 2 must therefore be included in the main document. We have also discussed this with the editorial team.

Point 2:

  1. Some phase 1 studies were included and the follow up period was not long enough. Robustness of data on survival might be limited.

Response 2:

We thank the reviewer for this important and critical comment. We again thoroughly reviewed the four studies (Chapman et al., 2018; Dhir et al., 2018; Miyasaka et al., 2019 and Tsujimoto et al., 2019) where the mOS was longer than the follow-up period. Qualitatively, the studies were robust and statistically presented correctly. The quality assessment of these studies was good to moderate as shown in suppl. table 2. We have made an additional comment to this in the discussion (limitations section, page 11).

Point 3:

  1. Please include data on adjuvant therapy after surgical resection in the table. The adjuvant therapy after NAC is clinically important. Can the authors suggest its relevance through this systematic review?

Response 3:

Data on impact of adjuvant therapy after NAC with GNP is scarce and no randomized trials exist, that compare patients with adjuvant therapy and those without in this setting. Available data about adjuvant therapy of included studies were added to table 2.

Point 4:

  1. Appropriate duration of NAC has not been fixed. Can the authors provide relationship of NAC duration and resection rates?

Response 4:

We thank the reviewer for this important enquiry. Overall, we believe that this question is best answered in a randomized controlled trial. A truly meaningful subgroup analysis cannot be achieved based on our data. Nevertheless, we would like to comment on this as follows.

Of the 21 studies, 4 studies did not specifically report the number of cycles of GNP. In the remaining studies, the number of median cycles varied from 2-8. Based on the clinical situation that a re-evaluation is carried out after 8 weeks (2 cycles of GNP, for FFX classically 4 cycles), we have formed 2 groups from this:

Resection rates all patients (R0+R1) - <=2 cycles (n=4): 0.54 (0.28, 0.80)

Resection rates all patients (R0+R1) - >2 cycles (n=10): 0.39 (0.20, 0.59)

This indicates that patients with 2 cycles have a resection rate of 54% compared to 39% of patients with more than 2 cycles of GNP.

However, many biases are present that limit the assessment of these data such as parallel RTx, primary decision 2 vs. 4 cycles by an MDT and dose or protocol variations.

For the latter, we have once again conducted subgroup analyses and see:

Resection rates all patients (R0+R1) – no protocol modification (n=11): 0.31 (0.17, 0.46)

Resection rates all patients (R0+R1) – with protocol modification (n=7): 0.40 (0.17, 0.66)

The fact that an improved resection rate is seen in the modified GNP protocol is also not simply explained and could again be due to further therapies such as RTx or subsequent system therapy (as reported in 4 studies see table 1).

Reviewer 2 Report

The authors have reported a systematic review and meta-analysis regarding the efficacy and safety of neoadjuvant Gemcitabine plus Nab-Paclitaxel in Borderline Resectable and Locally Advanced Pancreatic Cancer. They have demonstrated that this is the first systematic review and meta-analysis addressing outcomes of neoadjuvant Gemcitabine and Nab-Paclitaxel (GNP) in patients with BRPC or LAPC.

However, there are a few more recently published systematic reviews and meta-analysis data which are more elaborated and having comparison between FOLFIRINOX and gemcitabine plus nab-paclitaxel in the neoadjuvant chemotherapy, e.g.,

Janssen QP, O'Reilly EM, van Eijck CHJ and Groot Koerkamp B (2020) Neoadjuvant treatment in patients with resectable and borderline resectable pancreatic cancer. Front. Oncol. 10:41. doi: 10.3389/fonc.2020.00041;

Wolfe AR, Prabhakar D, Yildiz VO, Cloyd JM, Dillhoff M, Abushahin L, Alexandra Diaz D, Miller ED, Chen W, Frankel WL, Noonan A, Williams TM. Neoadjuvant-modified FOLFIRINOX vs nab-paclitaxel plus gemcitabine for borderline resectable or locally advanced pancreatic cancer patients who achieved surgical resection. Cancer Med. 2020 Jul;9(13):4711-4723. doi: 10.1002/cam4.3075.

Tang R, Meng Q, Wang W, Liang C, Hua J, Xu J, Yu X, Shi S. Head-to-head comparison between FOLFIRINOX and gemcitabine plus nab-paclitaxel in the neoadjuvant chemotherapy of localized pancreatic cancer: a systematic review and meta-analysis. Gland Surg 2021;10(5):1564-1575. doi: 10.21037/gs-21-16;

Even though the authors have considered studies until August 2020 a few of the required references are missing (as mentioned above in the first two references).

In conclusion, this systemic review and meta-analysis have tried to demonstrate the feasibility of neoadjuvant GNP in patients with BRPC or LAPC as a reasonable alternative use of FOLFIRINOX while required. However, the Forrest plot (Figure 2) is lacking FOLFIRINOX data. Forrest plot can further depict the difference in survival rate (e.g., 1- to 5-year) between patients who received FOLFIRINOX versus gemcitabine and nab-paclitaxel. Efficacy comparison of FOLFIRINOX versus gemcitabine and nab-paclitaxel can be represented by incorporating Kaplan-Meyer curve of Overall Survival. Prognostic Factors influencing the survival outcomes and Adverse Events can be included to refer toxicities of the treatments.

Therefore, this systemic review is lacking behind to match up the expectations of the readers of this journal.

Author Response

Reviewer 2:

The authors have reported a systematic review and meta-analysis regarding the efficacy and safety of neoadjuvant Gemcitabine plus Nab-Paclitaxel in Borderline Resectable and Locally Advanced Pancreatic Cancer. They have demonstrated that this is the first systematic review and meta-analysis addressing outcomes of neoadjuvant Gemcitabine and Nab-Paclitaxel (GNP) in patients with BRPC or LAPC.

Point 1:

However, there are a few more recently published systematic reviews and meta-analysis data which are more elaborated and having comparison between FOLFIRINOX and gemcitabine plus nab-paclitaxel in the neoadjuvant chemotherapy, e.g.,

Janssen QP, O'Reilly EM, van Eijck CHJ and Groot Koerkamp B (2020) Neoadjuvant treatment in patients with resectable and borderline resectable pancreatic cancer. Front. Oncol. 10:41. doi: 10.3389/fonc.2020.00041;

Response 1:

Compared to our work, only a systematic review and no meta-analysis was performed. Neoadjuvant treatment in resectable PDAC and BRPC was investigated. In contrast to our study, patients with LAPC were not included. According to the search terms of this systematic review (“neoadjuvant,” “FOLFIRINOX,” “folinic acid,” “fluorouracil,” “irinotecan,” “oxaliplatin,” “pancreas cancer,” “drug combination,” and relevant variants thereof) FOLFIRINOX was the drug combination of interest, not gemcitabine/ nab-paclitaxel. Table 3 provides an overview of recently published neoadjuvant trials from 2016 to 2019 (FOLFIRINOX vs. other than FOLFIRINOX). However, no trial about neoadjuvant GNP in BRPC was included. Especially no comparison between FOLFIRINOX and GNP was performed.

Point 2:

Wolfe AR, Prabhakar D, Yildiz VO, Cloyd JM, Dillhoff M, Abushahin L, Alexandra Diaz D, Miller ED, Chen W, Frankel WL, Noonan A, Williams TM. Neoadjuvant-modified FOLFIRINOX vs nab-paclitaxel plus gemcitabine for borderline resectable or locally advanced pancreatic cancer patients who achieved surgical resection. Cancer Med. 2020 Jul;9(13):4711-4723. doi: 10.1002/cam4.3075.

Response 2:

This is no systematic review or meta-analysis. In this retrospective study, only resected patients (52 received FOLFIRINOX and 20 received GNP) were included. This is no intention-to-treat analysis like studies included in our Meta Analysis. The primary outcome investigated was survival of resected patients and not resection rate.

Point 3:

Tang R, Meng Q, Wang W, Liang C, Hua J, Xu J, Yu X, Shi S. Head-to-head comparison between FOLFIRINOX and gemcitabine plus nab-paclitaxel in the neoadjuvant chemotherapy of localized pancreatic cancer: a systematic review and meta-analysis. Gland Surg 2021;10(5):1564-1575. doi: 10.21037/gs-21-16;

Reponse 3:

Only 8 studies were included in this meta-analysis. The primary outcome was survival, not resection rate. Also conference abstracts were included (e.g. Gage et al., 2019), which probably affects the quality of the data. For this reason, conference abstracts were excluded from our analysis. This meta-analysis also includes patients with resectable PDAC; these patients were excluded from our analysis. In addition, a meta-analysis of retrospective studies as head to head comparison of therapeutic regimes should be interpreted cautiously because of the risk of bias.

In summary, the subject of our work is clearly different and offers significant added value to the mentioned studies.

Point 4:

Even though the authors have considered studies until August 2020 a few of the required references are missing (as mentioned above in the first two references). In conclusion, this systemic review and meta-analysis have tried to demonstrate the feasibility of neoadjuvant GNP in patients with BRPC or LAPC as a reasonable alternative use of FOLFIRINOX while required. However, the Forrest plot (Figure 2) is lacking FOLFIRINOX data. Forrest plot can further depict the difference in survival rate (e.g., 1- to 5-year) between patients who received FOLFIRINOX versus gemcitabine and nab-paclitaxel. Efficacy comparison of FOLFIRINOX versus gemcitabine and nab-paclitaxel can be represented by incorporating Kaplan-Meyer curve of Overall Survival. Prognostic Factors influencing the survival outcomes and Adverse Events can be included to refer toxicities of the treatments.

Response 4:

The primary endpoint of our work was to provide pooled resection rates after neoadjuvant treatment with GNP. The inclusion of patients treated with FOLFIRINOX or the comparison of GNP with FOLFIRINOX was not intended, as no randomized trials were available at the time of the literature search. Patient- level survival data to compute Kaplan-Meyer curves was not available and was not in our research scope for the question we wanted to answer.

Reviewer 3 Report

This is a clearly presented study. Some minor comments for its improvement:

a) As there is high heterogeneity in the studies, it would be nice if the authors mentioned the steps taken to investigate potential sources of it (e.g. sensitivity analysis or meta-regression)

b) Did the GNP protocol (standard vs modified) or the number of cycles have any impact on the results?

c) Consider renumbering the Tables (Table 1, 2, S1, S2), as all of them are in the Supplemental file and could be confusing to the reader

Thanks a lot for your contribution

Author Response

Reviewer 3:

Point 1:

This is a clearly presented study. Some minor comments for its improvement:

Response 1:

We thank the expert for this positive assessment.

Point 2:

  1. a) As there is high heterogeneity in the studies, it would be nice if the authors mentioned the steps taken to investigate potential sources of it (e.g. sensitivity analysis or meta-regression)

Response 2:

We have identified that according to the I2 statistic we have high heterogeneity between studies. We have tried stratifying by using various subgroup analyses to investigate the sources for this. We performed one analysis based on LAPC and BRPC, another one based on R0 resection rates only, and in the article’s supplement we have provided pooled resection rates from prospective studies only. Unfortunately, none of these subgroups analyses managed to lower the I2 statistic below 50%. We can only hypothesize which additional factors are responsible. Most of the cross-sectional studies were classified as having medium study quality, while the prospective studies all had high or unclear risk of bias. We suspect that these methodological differences and less robust methodological implementation is the main source of the heterogeneity. However, due to the clinical relevance of the treatment for this specific patient population, we believe our study results can guide informed decision on both physician’s and patient’s side on what is the best treatment option, until further studies, such as RCTs, become available.

Point 3:

  1. b) Did the GNP protocol (standard vs modified) or the number of cycles have any impact on the results?

Response 3:

We thank the reviewer for this important request. We have already commented on this in considerable detail in reviewer 1's answer 4 and would like to refer to this.

Point 4:

  1. c) Consider renumbering the Tables (Table 1, 2, S1, S2), as all of them are in the Supplemental file and could be confusing to the reader

Response:

This is an important point that we have addressed before. Since tables 1 and 2 could not be inserted directly into the main document, they were accidentally transferred to the supplement. We have tried to change this in the renewed submission.

Thanks a lot for your contribution

Round 2

Reviewer 2 Report

The present form of the Article includes meta-analysis to demonstrate – ‘the feasibility of neoadjuvant GNP in patients with BRPC or LAPC, representing a reasonable alternative in this setting, when comorbidities preclude the use of FOLFIRINOX’. Yes, it is important to investigate the resection rates after GNP treatment, however to demonstrate the GNP treatment as a very important alternative of FOLFIRINOX authors must include comparison group of FOLFIRINOX as suggested previously.

The references suggested before are proposing the importance of incorporating the comparison studies of GNP with FOLFIRINOX whether the primary outcome is survival or resection rate.

Also, authors are suggested to include the reference of the systematic review which was already performed.

Author Response

We would like to comment again on the remarks made by reviewer 2:

Point 1: The present form of the Article includes meta-analysis to demonstrate – ‘the feasibility of neoadjuvant GNP in patients with BRPC or LAPC, representing a reasonable alternative in this setting, when comorbidities preclude the use of FOLFIRINOX’. Yes, it is important to investigate the resection rates after GNP treatment, however to demonstrate the GNP treatment as a very important alternative of FOLFIRINOX authors must include comparison group of FOLFIRINOX as suggested previously.
Response 1: The primary endpoint of our work was to provide pooled resection rates after neoadjuvant treatment with GNP. We registered our work in PROSPERA. The inclusion of patients treated with FOLFIRINOX or the comparison of GNP with FOLFIRINOX was not intended, as no randomized trials were available at the time of the literature search. Data for FOLFIRINOX treatment in LAPC and BRPC already exist and have been published e.g. by Petrilli et al. 2015 Pancreas, Suker et al. 2016 Lancet Oncology, Janssen et al. 2019 (JNCI J Natl Cancer Inst) and Chen et al. 2021 Medicine. In the discussion, we compared and interpreted our data with the NEOLAP study by Kunzmann et al. In this study, for the first time a group of LAPC patients receiving GNP or mFOLFIRINOX was investigated. There was no significant difference in resection rate and overall survival. Since patients differ in terms of their clinical characteristics under non-randomized conditions, as already described in detail in the last rebuttal letter, a direct comparison is not meaningful. A statistically valid, randomised and prospective study should address this question.

Point 2: The references suggested before are proposing the importance of incorporating the comparison studies of GNP with FOLFIRINOX whether the primary outcome is survival or resection rate. Also, authors are suggested to include the reference of the systematic review which was already performed.
Response 2: We thank the reviewer for his renewed objection. In the last rebuttal letter, we commented on the proposed study in great detail. The studies mentioned all have major limitations and cannot be used as comparative studies (e.g. different target cohort, retrospective, different endpoint). In our discussion, we have very carefully discussed our findings with the current literature (Janssen et al. 2019, Chen et al. 2021, Suker et al. 2016, Xu et al. 2019)

Round 3

Reviewer 2 Report

I'm Ok to refer the response from Academic editors.